# Reduction of Doxorubicin-Induced Cardiotoxicity by Co-Administration of Smart Liposomal Doxorubicin and Free Quercetin: In Vitro and In Vivo Studies

**DOI:** 10.3390/pharmaceutics15071920

**Published:** 2023-07-11

**Authors:** Hamidreza Dorostkar, Bibi Fatemeh Haghiralsadat, Mahdie Hemati, Fatemeh Safari, Azam Hassanpour, Seyed Morteza Naghib, Mohammad Hossein Roozbahani, M. R. Mozafari, Ali Moradi

**Affiliations:** 1Department of Clinical Biochemistry, Faculty of Medicine, Shahid Sadoughi University of Medical Sciences, Yazd 8916877391, Iran; hr.dorostkar@yahoo.com (H.D.); m.hemati1420@gmail.com (M.H.); 2Department of Advanced Medical Sciences and Technologies, Faculty of Paramedicine, Shahid Sadoughi University of Medical Sciences, Yazd 8916877391, Iran; fhaghirosadat@gmail.com; 3Medical Nanotechnology and Tissue Engineering Research Center, Yazd Reproductive Sciences Institute, Shahid Sadoughi University of Medical Sciences, Yazd 8916877391, Iran; 4Department of Physiology, Faculty of Medicine, Shahid Sadoughi University of Medical Sciences, Yazd 8916877391, Iran; fa.cardio@gmail.com; 5Cardiovascular Research Center, Shahid Sadoughi University of Medical Sciences, Yazd 8916877391, Iran; 6Department of Anatomical Sciences, Faculty of Medicine, Shahid Sadoughi University of Medical Sciences, Yazd 8916877391, Iran; azamhassanpour63@yahoo.com; 7Nanotechnology Department, School of Advanced Technologies, Iran University of Science and Technology and Biomaterials and Tissue Engineering Department, Breast Cancer Research Center, Motamed Cancer Institute, IUST, ACECR, Tehran 1684613114, Iran; naghib@iust.ac.ir; 8School of Advanced Technologies, Iran University of Science & Technology, Tehran 1684613114, Iran; 9Australasian Nanoscience and Nanotechnology Initiative (ANNI), Monash University LPO, Clayton, VIC 3168, Australia; dr.m.r.mozafari@gmail.com

**Keywords:** antioxidants, cardiotoxicity, doxorubicin, liposomes, oxidative stress, quercetin

## Abstract

Doxorubicin is one of the most effective chemotherapeutic agents; however, it has various side effects, such as cardiotoxicity. Therefore, novel methods are needed to reduce its adverse effects. Quercetin is a natural flavonoid with many biological activities. Liposomes are lipid-based carriers widely used in medicine for drug delivery. In this study, liposomal doxorubicin with favorable characteristics was designed and synthesized by the thin-film method, and its physicochemical properties were investigated by different laboratory techniques. Then, the impact of the carrier, empty liposomes, free doxorubicin, liposomal doxorubicin, and quercetin were analyzed in animal models. To evaluate the interventions, measurements of cardiac enzymes, oxidative stress and antioxidant markers, and protein expression were performed, as well as histopathological studies. Additionally, cytotoxicity assay and cellular uptake were carried out on H9c2 cells. The mean size of the designed liposomes was 98.8 nm, and the encapsulation efficiency (EE%) was about 85%. The designed liposomes were anionic and pH-sensitive and had a controlled release pattern with excellent stability. Co-administration of liposomal doxorubicin with free quercetin to rats led to decreased weight loss, creatine kinase (CK-MB), lactate dehydrogenase (LDH), and malondialdehyde (MDA), while it increased the activity of glutathione peroxidase, catalase, and superoxide dismutase enzymes in their left ventricles. Additionally, it changed the expression of NOX1, Rac1, Rac1-GTP, SIRT3, and Bcl-2 proteins, and caused tissue injury and cell cytotoxicity. Our data showed that interventions can increase antioxidant capacity, reduce oxidative stress and apoptosis in heart tissue, and lead to fewer complications. Overall, the use of liposomal doxorubicin alone or the co-administration of free doxorubicin with free quercetin showed promising results.

## 1. Introduction

The incidence of cancer has increased due to population growth and aging, as well as the increasing prevalence of risk factors, such as smoking, being overweight, and a Western lifestyle [1]. At the beginning of 2019, more than 16.9 million Americans had a history of cancer, and this number is expected to reach more than 22.1 million by the beginning of 2030 [2]; thus, cancer treatment has remained a serious challenge, as it has for centuries [3].

Adriamycin is one of the most effective chemotherapeutic drugs widely used against leukemia, lymphoma, and many solid tumors [4]. The high effectiveness of this drug, despite the dose-limiting toxicities, has attracted the attention of many researchers [5]. The commercial name of this anthracycline antibiotic is doxorubicin (DOX). Different mechanisms of action have been proposed for this drug, including a reaction with iron, a disruption of calcium homeostasis, changes in the activity of intracellular or mitochondrial oxidant enzymes, and a binding to topoisomerases [6]. It has been shown that numerous tissues are sensitive to the side effects of doxorubicin. Extensive damage to cardiomyocytes is generally irreversible because they are not capable of dividing again, so such an injury could affect cardiac function. Chronic administration of doxorubicin causes dose-dependent and irreparable cardiac toxicity, as well as arrhythmias that limit its use [4]. Cardiotoxicity is the main problem of treatment with this powerful chemotherapy drug, which is also the first line in the treatment of breast cancer [7]. Acute cardiotoxicity is mainly associated with increased inflammation, which can lead to myocardial and pericardial inflammation syndrome, which is usually reversible upon discontinuation of the drug. Cardiomyopathy is the most common adverse effect of anthracycline cardiotoxicity and can occur months or years following exposure to these types of compounds. Anthracycline-induced cardiotoxicity is a dose-dependent toxicity characterized by cardiac cell death and can result in left ventricular dysfunction and symptomatic heart failure [8]. Molecular mechanisms of cardiotoxicity include oxidative stress, such as that caused by mitochondrial-dependent reactive oxygen species (ROS), nitrogen oxygen species (NOX)-dependent ROS, NAD(P)H-dependent ROS, apoptosis, changes in energy homeostasis, and other mechanisms [9]. The use of chemotherapy is very limited due to the emergence of side effects in patients; therefore, the development of safe, natural, and effective methods with less toxicity or adverse effects is essential. The possibility of treatment with herbal substances, especially flavonoids, has raised this issue more. Flavonoids are found in many fruits and vegetables and have functional compounds with a phenylbenzopyrene structure [10]. Pentahydroxyflavone (C15H10O7), also called quercetin, is a flavonoid present in plants and foods such as onions, green tea, and broccoli. It is a powerful antioxidant and performs myriad biological activities, such as providing protection in ischemic reperfusion myocardial injury, reducing oxidative stress by mechanisms such as the inhibition of NADPH oxidase, inhibition of xanthine oxidase, activation of selenoproteins, and suppression of the Fenton reaction, and has anti-inflammatory, anti-apoptotic, and anti-tumor properties. It is also able to scavenge ROS directly as a result of its specific flavonoid structure or indirectly owing to the restoration of intrinsic antioxidant systems [11,12].

Nanomaterials used in medicine include lipid-based carriers, polymeric carriers, metal nanoparticles, and carbon-structured and mineral particles that have multiple functions and are used for different purposes and as carriers of a variety of substances, such as drugs [13]. Several strategies have been used to increase the efficacy and safety profile of DOX, such as the use of liposomes [5]. The first nano drug delivery system in clinical use was liposomes, and the first approved drug nanocarrier was the PEGylated liposomal form of doxorubicin under the brand name Doxil [14]. Liposomes possess a closed phospholipid bilayer membrane system that has attracted much attention over the last 30 years as a drug delivery system with high potential. Their ability to package both hydrophilic and hydrophobic drugs, along with their biocompatibility and biodegradation, has made them useful carriers for drug delivery. In addition, excellent technical advances in this field have led to their widespread use in various applications as anti-cancer drug delivery systems and bioactive molecules, as well as in diagnostics, and as therapeutic agents [15].

Since the use of chemotherapeutic drugs, such as doxorubicin, is inevitable in the treatment of cancer, and considering their serious side effects on the health of the human body, seeking new therapies with less cytotoxicity is required. To this aim, anionic liposomal nanocarriers, with controlled release patterns and high sensitivity to an acidic environment, were designed and synthesized to deliver doxorubicin. Liposomal doxorubicin is easily manufactured and has suitable characteristics to be applied as a nanocarrier. Furthermore, quercetin was employed as an herbal antioxidant in combination with encapsulated doxorubicin (co-administration) to evaluate a possible beneficial effect of these two compounds in reducing the side effects of doxorubicin. The experimental procedures were continued in a rat model (in vivo), along with the H9c2 cell line in vitro. The cytotoxicity of doxorubicin alone and its encapsulated form in combination with quercetin was evaluated using biochemical analyses, antioxidant and oxidative stress markers, apoptosis assay, and histopathological assessments (Figure 1).

## 2. Materials and Methods

### 2.1. Materials

Doxorubicin HCl (DOX) and QC (purity > 95%) were obtained from Ebewe Pharma (Unterach, Austria) and Sigma-Aldrich (St. Louis, MO, USA), respectively. Soybean phosphatidylcholine and DSPE-PEG2000 (distearoyl phosphoethanolamine-polyethylene glycol) were obtained from Lipoid GmbH (Ludwigshafen, Germany). Cholesterol was supplied by Sigma-Aldrich (St. Louis, MO, USA). PBS tablets, dialysis bags (MW  =  12  kDa), DMSO (dimethyl sulfoxide), MTT (3–(4,5-dimethylthiazol-2-yl)-2,5-diphenyl tetrazolium bromide) and paraformaldehyde solution were procured from Sigma-Aldrich (St. Louis, MO). DAPI (4′,6-diamidino2-phenylindole) and DIL Stain (1,1′-Dioctadecyl-3,3,3′,3′-Tetramethylindocarbocyanine Perchlorate) were supplied by Thermo Fisher Scientific (Waltham, MA, USA). All other chemicals, solvents, and salts were of the analytical grade and used without further purification unless specified.

### 2.2. Design and Synthesis of Nanocarriers

In this study, liposomes were synthesized using the thin-film hydration method. Soybean phosphatidylcholine (SPC) (Lipoid GmbH, Ludwigshafen, Germany), cholesterol (Sigma, St. Louis, MO, USA), and distearoyl phosphoethanolamine-polyethylene glycol 2000 (DSPE-mPEG) (Lipoid GmbH, Ludwigshafen, Germany) were weighed at a molar ratio of 66.5:28.5:5. They were dissolved in chloroform and placed in a round-bottom flask. The thin film was formed after evaporation of the solvent in a rotary instrument (Heidolph, Schwabach, Germany). Doxorubicin (Ebewe Pharma, Unterach. Austria) at a concentration of 0.5 mg/mL in phosphate-buffered saline (PBS; pH 7.4) was prepared and added to a flask containing the lipid film. The film was hydrated at 55 °C for 60 min. In this stage, the drug was inactively encapsulated in multi-lamellar vesicles (MLVs). To synthesize empty liposomes, the hydration step was conducted with PBS only. Probe sonication (E–Chrom Tech Co., Taipei, Taiwan) was used to reduce the size of the MLVs, form monolayer liposomes or single uni-lamellar vesicles (SUV), and homogenize them. The resulting nanocarriers were passed through 0.45 μm and 0.22 μm filters, and membrane dialysis was applied to release drugs that were not incorporated into liposomes. In this method, the synthesized nanocarriers were placed in a dialysis bag (cutoff = 12–14 kDa) and a container containing PBS at 4 °C and pH 7.4. While the solution containing nanocarriers was stirred, free drugs were passed through the dialysis bag and flowed into the PBS, and thus removed. Finally, the drug carrier (encapsulated doxorubicin) was stored away from light at 2–8 °C.

### 2.3. Characterization and Stability of Nanocarriers

Particle size, polydispersity index (PDI) (a measure of the heterogeneity of a sample based on size), and zeta potential (electric charge) of the synthesized nanocarriers were measured by a Zeta-sizer device using the dynamic laser scattering (DLS) technique (Brookhaven Corp., Holtsville, NY, USA). The shapes of nanocarriers were analyzed by atomic force microscopy (AFM) (NanoWizard, Berlin, Germany). The Fourier-transform infrared (FTIR) spectroscopy method (Shimadzu, Tokyo, Japan) was used to assess the molecular interactions between the drug and liposomes. The field emission scanning electron microscope (FE-SEM) (TESCAN, Brno, Czechia) was applied to observe the morphology of the synthesized nanoparticles. The stability of the encapsulated form of doxorubicin was evaluated by analyzing the changes in size, PDI, and zeta potential of liposomes after 3 months of storage at 2–8 °C and away from light. To determine the percentage of drug encapsulation efficiency (EE%), liposomes were lysed with isopropanol (Merck, Darmstadt, Germany), and their optical absorption was determined by spectrophotometry (Perkin Elmer, Ueberlingen, Germany). The concentration of doxorubicin was calculated according to the standard curve. The percentage of EE was equal to the quantity of loaded drugs divided by the quantity of consumed drugs, multiplied by 100.

### 2.4. In Vitro Drug Release

The release of liposomal and free doxorubicin was evaluated in four different settings using a dialysis bag (cutoff = 12–14 kDa) containing PBS with pH 7.4 and pH 5.4 at 37 °C and 42 °C. Briefly, the dialysis bag containing drug/liposomes was placed in a beaker containing PBS and a magnetic stirrer. After 1, 2, 4, 8, 24, 48, and 72 h, samples were taken and immediately replaced with fresh PBS to maintain the sink condition throughout the experiment. The amount of the drug in different samples was measured by the spectrophotometric technique according to the standard drug curve.

### 2.5. Animal Groups and Treatment

Forty-two male Wistar rats (150–180 g) were purchased from the animal laboratory of the Shahid Sadoughi University of Medical Sciences, Yazd, Iran. The animals had free access to water and food and were maintained at a suitable temperature and humidity during a 12 h dark and light cycle. All experimental procedures in this study were performed according to the ethical guidelines for animal research approved by the Ethics Committee for Animal Experiments of Shahid Sadoughi University of Medical Sciences, Yazd, Iran.

The treatment period lasted for 12 days. The weight of rats was measured on the first and last day with a scale (A&D, Tokyo, Japan). Doxorubicin was selected as a common chemotherapy drug [16,17,18,19], and a cumulative dose of 18 mg/kg was used in 6 injections every other day via intraperitoneal injection (IP). Due to the antioxidant role of quercetin [20,21,22,23,24], this herbal substance (purity > 95%) (Sigma, USA) was selected and administered daily by gavage at a dose of 50 mg/kg with a normal saline carrier for 11 days. Rats were randomly divided into seven groups of six members as follows. Vehicle (V): animals receiving normal saline by gavage and PBS via IP; quercetin (Q): quercetin by gavage and PBS via IP; blank liposome (L): normal saline by gavage and empty liposomes via IP; doxorubicin (D): normal saline by gavage and free doxorubicin via IP; liposomal doxorubicin (LD): normal saline by gavage and liposomal doxorubicin via IP; doxorubicin and quercetin (DQ): quercetin by gavage and free doxorubicin via IP; LDQ or liposomal doxorubicin and quercetin (LDQ): quercetin by gavage and liposomal doxorubicin via IP (Table 1). On the twelfth day, after anesthesia with ketamine (90 mg/kg) and xylazine (10 mg/kg), fasting blood samples were taken from rats. After blood coagulation, serum was separated by centrifugation at 3000 *g* for 15 min. The cardiac tissue was immediately removed and weighed after washing with cold normal saline, and the left ventricular tissue was stored at −70 °C for further experiments. Some cardiac tissues were also placed in formalin for histopathological evaluation.

### 2.6. Biochemical Assessments

To assess cardiac tissue damage, serum creatine phosphokinase-MB (CK-MB) and lactate dehydrogenase (LDH) were measured using commercial kits (Pars Azmoun, Iran). The Bradford technique was used following homogenization to determine the protein level of the left ventricle [25].

### 2.7. Analyzing the Activity of Antioxidant Enzymes and the Levels of Oxidative Stress

To analyze the antioxidant capacity of left ventricular tissue, the activity levels of superoxide dismutase (SOD), catalase (Cat), and glutathione peroxidase (GPX) enzymes were measured by the spectrophotometric technique using commercially available kits (Zellbio, Lonsee, Germany). Oxidative stress was measured in the same way by a commercial kit designed to measure malondialdehyde (MDA) (Zellbio, Lonsee, Germany).

### 2.8. Western Blot Analysis

The expression levels of NADPH oxidase 1 (NOX1), Ras-related C3 botulinum toxin substrate 1 (Rac1), Rac1-GTP, and sirtuin-3 (SIRT3) proteins, involved in oxidative stress and antioxidant status, as well as the expression of B-cell lymphoma-2 (Bcl-2), involved in apoptosis, were evaluated in the left ventricle of the rats by the Western blot technique. In summary, 40 mg of the left ventricular tissue was homogenized in cold PBS containing the protease inhibitor. In the next step, after centrifugation (15,000 *g*, 4 °C, 30 min), the protein contents of the resulting supernatant were measured by the Bradford method. The extracted proteins (50 μg) were run on a polyacrylamide gel containing 10% sodium dodecyl sulfate to separate them in terms of their molecular weight and then transferred to the nitrocellulose membrane. In the blocking stage, 5% bovine serum albumin was used for 2 h at room temperature. In the next stage, the membranes were incubated overnight with primary antibodies against NOX1 (Abcam, Cambridge, MA, USA), Rac1 (Abcam, USA), Rac1-GTP (NewEast Biosciences, Malver, PA, USA), SIRT3 (Abcam, USA), Bcl-2 (Santa Cruz Biotechnology, Cambridge, MA, USA), and beta-actin (Abcam, Cambridge, MA, USA). Afterward, secondary antibodies conjugated with horseradish peroxidase (Cell Signaling Co., Munich, Germany) were applied and incubated for 2 h. In the final step, the protein bands were visualized using an enhanced chemiluminescence kit (GE Healthcare, Bucks, UK) and then semi-quantified by Image J software. The expression levels of all proteins were normalized against the beta-actin protein as an internal control.

### 2.9. Histopathological Evaluations

As mentioned earlier, some fresh cardiac tissues were placed in 10% formalin for histopathological examinations. Tissues were paraffin-embedded and then sectioned at a thickness of 5 μm to prepare slides. Hematoxylin-eosin (HE) staining was performed to evaluate the degree of injury in cardiac tissue. Finally, the prepared slides were observed under a light microscope. Additionally, histopathological images were taken from the left ventricular tissues.

### 2.10. In Vitro Cellular Uptake

The fluorescence intensity was observed to evaluate the distribution of doxorubicin and quercetin in the cardiac cell line. Briefly, H9c2 cells (Pasteur Institute, Tehran, Iran) were seeded onto 6-well plates (1.5 × 10^5^ cells per well). In the next step, they were treated with blank-DiL (1,1′-Dioctadecyl-3,3,3′,3′-Tetramethylindocarbocyanine Perchlorate) (Thermo Fisher Scientific, USA) liposome, doxorubicin (30 µg mL^−1^), liposomal doxorubicin (30 µg mL^−1^), quercetin (300 µg mL^−1^), doxorubicin and quercetin, liposomal-doxorubicin plus free quercetin. After incubation (3 h), cells were washed twice with PBS (pH 7.4) and then fixed with the ethanol solution (95%). The nuclei of cells were counterstained with DAPI (4′,6-diamidino2-phenylindole) (Thermo Fisher Scientific, USA) at a concentration of 1 mg mL^−1^ for 15 min. Finally, the images were acquired by fluorescence microscopy (Olympus, Tokyo, Japan).

### 2.11. Cytotoxicity Study

The MTT (3–(4,5-dimethylthiazol-2-yl)-2,5-diphenyl tetrazolium bromide) (Sigma-Aldrich, St. Louis, MO, USA) assay was conducted to analyze the cytotoxicity of the prepared liposomes, doxorubicin, and quercetin. For this purpose, 10^4^ cells were treated with different concentrations of quercetin (12.5, 25, 50, 100, and 150 µg mL^−1^), doxorubicin, and liposomal doxorubicin (1.25, 2.5, 5, 10, and 15 µg mL^−1^) in a 96-well plate for 48 h. Next, the contents of the wells were removed and incubated with 10 µL of MTT (5 mg mL^−1^) and 90 µL of the cell culture medium for 3 h. The resulting formazan crystals were dissolved in dimethyl sulfoxide (DMSO) (Sigma-Aldrich, USA). The absorbance of samples was measured (570 nm) by a microplate spectrophotometer (BioTek Synergy HTX, Winooski, VT, USA).

### 2.12. Statistical Analysis

Graphpad Prism software (version 6) was used for the statistical analysis. Quantitative results were reported as the means and standard deviations (mean ± SD). One-way analysis of variance (ANOVA) followed by Tukey’s post hoc test were applied for multiple comparisons. The *p*-value of less than 0.05 was considered statistically significant.

Statement: we confirm that all methods in our study are reported in accordance with the ARRIVE guidelines.

## 3. Results and Discussion

### 3.1. Characteristics and Stability of Nanocarriers

Liposomes formulated in this study had suitable features, such as a simple synthesis method, anionic charge, controlled release pattern, sensitivity to pH, suitable size, acceptable PDI, and stability, with a high EE percentage (85% ± 1.4). Studies have shown that anionic lipid vesicles are less toxic and better tolerated by cells and tissues in vitro and in vivo [26,27]. The characteristics of empty and liposome-loaded doxorubicin, as well as doxorubicin-containing liposomes stored for 3 months after synthesis, are shown in Table 2. There were no significant differences in the physicochemical properties of empty liposomes, newly synthesized liposome-loaded doxorubicin, and doxorubicin-containing liposomes stored for 3 months.

The size and shape of doxorubicin-containing liposomes were confirmed by the AFM technique (Figure 2a). Doxil was synthesized with the same ingredients as a carrier, butwith different methods and ratios. Haghiralsadat et al. employed 1,2 dipalmitoyl-sn-glycero-3-phosphocholine (DPPC) instead of soybean phosphatidylcholine (SPC) using a pH gradient method and different proportions of materials. They synthesized liposomes with slightly more negative charges but with sizes and EE percentages close to the liposomes of the present study. Choosing a suitable lipid for the synthesis of liposomes is essential because the stability of liposomes depends on the stability of their lipid bilayer. Various methods and compounds have been reported in studies for the synthesis of liposomes [14,28,29]. Park et al. developed doxorubicin-loaded liposomal iron oxide nanoparticles and evaluated their application in vitro and in vivo for combined chemo-photothermal cancer therapy. Finally, after studying the initial efficacy, utility, and safety of their nanosystem, they reported that it may serve as an effective and safe agent for combination cancer therapy [30]. Electron microscopy images showed a proper morphology of the spherical and smooth liposomes (Figure 2b). FTIR analysis is used to detect chemical and thermodynamic changes in structures of biological systems. The results of FTIR (Figure 2c) were similar to previous studies and further confirmed the suitability of nanocarriers. The FTIR results of empty liposomes and liposomal doxorubicin corroborate the lack of chemical interactions between doxorubicin and liposomes, as well as the stability of doxorubicin within liposomes, as observed in the present study [28,31].

The 3-month stability of the synthesized liposomes was evaluated by examining the characteristics of fresh and stored liposomes and was reported as appropriate. Haghiralsadat and colleagues considered such similarity to indicate liposome stability, which could be a step towards reducing the problem of short liposome stability [28]. Ferreira et al. reported that the storage of liposomes could result in an increase in their size due to aggregation or fusion, which might lead to their rapid uptake by phagocytes. This phenomenon can reduce the half-life of liposomes. Fusion of vesicles also results in carrier leakage and changes in surface charge. Ferreira et al. attributed the stability of the carrier size over time to the presence of a high negative charge on the liposome surface and the existence of PEG branches, which prevent aggregation and enhance carrier stability [32].

### 3.2. Evaluation of Drug Release

The cumulative release percentages of liposomal nanocarriers and the free drug in four different conditions, at two temperatures and two pH values, were assessed and compared (Figure 3). Similar interesting behavior was detected in some release conditions. Two remarkable features of existing anionic liposomes were that they possess controlled release patterns and pH sensitivity. The free drug reached its peak release after 8 h; however, this time was longer for the synthesized liposomes (24 h). The release peak of the free drug in different conditions was 25% to 37.5% higher than the liposome release peak in the same conditions, which indicates the controlled release of the synthesized liposomes. Further, a 20% increase in the release peak of liposomes in an acidic medium compared to neutral pH implies the pH sensitivity of liposomes that could serve for passive targeting. This ability causes less damage to healthy tissues and has a greater effect on target tissues [28]. Encapsulation of DOX in pH-sensitive liposomes has been reported to improve DOX accumulation in tumors [5]. Biabanikhankahdani and colleagues reported that doxorubicin becomes more hydrophilic at a low pH, and its solubility in water is increased. Since the surrounding environment of tumor tissues, lysosomes, and intracellular endosomes are also acidic, doxorubicin is released after being taken up by these tissues. Sensitivity to pH is an effective property for delivering anti-cancer compounds to tumors because it keeps these compounds at the physiological pH and increases the diffusion rate in the tumor tissue environment [33]. Chen and co-workers reported that the pH sensitivity may be due to PEG inhibiting the release of doxorubicin at the physiological pH. With increasing acidity, long PEG chains are gradually separated, and this may be one of the possible reasons for the increasing drug release [34]. Duarte et al. used a newly formulated, pH-sensitive liposome capable of encapsulating DOX and Simvastatin to evaluate its therapeutic potential for breast tumors. Finally, they observed the inhibitory effect on breast cancer cell lines in both free and encapsulated drugs and reported a promising potential of encapsulated liposomes for breast tumor treatment [7].

### 3.3. Animal Studies

A comparison of the weight change of rats at the initiation and end of the intervention and the ratio of heart to body weight at the end of the study was made (Figure 4). One of the reasons for studying nanoparticles for chemotherapeutic purposes is that they can lead to toxicity in healthy tissues, and therefore need to be investigated [32]. In the present study, the synthesized empty liposome did not show a negative effect on the analyzed parameters in animals. This may indicate a lack of observable toxicity or interference of liposome blanks. In the present study, free doxorubicin led to weight loss in rats. This weight loss was significantly improved in the groups that used the liposomal form, quercetin, or both of these interventions. Co-administration of liposomal doxorubicin with quercetin exhibited the most significant effect on improving doxorubicin-induced weight loss in animals. Pouna and co-workers attributed the doxorubicin-induced weight loss to the general toxicity of anti-cancer agents [18]. Our findings demonstrated that the ratio of the heart weight to the total body weight was not significantly different in the doxorubicin-treated group compared with the normal (vehicle) group. Shaker et al. stated that this phenomenon may be owing to the simultaneous body weight loss in the doxorubicin-treated group [35]. Zakaria et al. found that the administration of quercetin before, and, similarly to the present study, during, the treatment course led to reduced doxorubicin-induced weight loss [20].

### 3.4. Evaluation of Biochemical, Antioxidant, and Oxidative Stress Status

The results of the analysis of biochemical, antioxidant capacity, and oxidative stress markers are represented in Figure 5. Doxorubicin increased cardiac-specific enzymes in the studied groups. Co-administration of liposomal doxorubicin and quercetin showed better effects on the cardiac-specific enzymes compared with the use of doxorubicin alone, liposomal doxorubicin alone, and doxorubicin plus quercetin (Figure 5A,B). Qureshi et al. evaluated the impact of mPEG-PLGA copolymer on cardiac-related biochemical markers and observed that the specific markers were increased in the doxorubicin-treated group, whereas they were decreased in the encapsulated doxorubicin-treated and the doxorubicin+quercetin groups. In agreement with our findings, they also observed a further reduction in the cardiac-specific markers when encapsulated doxorubicin and quercetin were simultaneously administrated [36]. Paliwal et al. reported a decrease in cardiac enzymes after subjects received liposomal doxorubicin, denoting the lower toxicity of the encapsulated form of the drug. Therefore, they applied this strategy to the intracellular delivery of anti-cancer drugs [37]. Studies have shown that quercetin has antioxidant and free radical scavenging properties that may help protect the myocardium and inhibit cardiac enzyme leakage [20,38], which was confirmed in our study.

The free form of doxorubicin increased MDA levels in the cardiac tissues of animals. It was shown that the simultaneous administration of liposomal doxorubicin and quercetin has the greatest reducing effect on this lipid peroxidation index (Figure 5C). The first target of doxorubicin-induced free radical damage was lipid-rich cell membranes. Increased lipid peroxidation is a biochemical marker for oxidative stress in doxorubicin-treated rats [38]. In the present study, doxorubicin reduced the activity of glutathione peroxidase (Figure 5D), catalase (Figure 5E), and superoxide dismutase enzymes (Figure 5F) in cardiac tissues. Decreased activity of antioxidant enzymes in the cardiac tissues of doxorubicin-treated rats was also addressed in other studies. This is related to the possibility of superoxide radical production, which affected the production and activity of enzymes and led to the accumulation of these radicals in the myocardium [38]. In this regard, Nazmi et al. reported that doxorubicin reduces the antioxidant capacity in the cardiac tissue by reducing the activity of catalase, which decreases the amount of hydrogen peroxide radicals. The activation of myocardial antioxidants is a promising topic in the treatment of cardiovascular disorders caused by increased oxidative stress, and it has been shown that pre-treatment with quercetin can reduce the side effects of this drug [38,39].

### 3.5. Expression of Proteins Involved in Oxidative Stress and Apoptosis

The results of the Western blot analysis are shown in Figure 6A (additional information are available in the Appendix A). Doxorubicin increased the expression of NOX1, Rac1, and Rac1-GTP proteins in left ventricular tissue (Figure 6B). It also decreased the expression of SIRT3 and Bcl-2 proteins (Figure 6C); however, co-administration of liposomal doxorubicin and quercetin attenuated the changes in the expression of the above genes induced by doxorubicin (Figure 6A–C). Similarly to our findings, Iwata et al. reported that the administration of doxorubicin to mice resulted in increased NOX1 expression in the cardiac tissue, which was associated with increased ROS levels. They identified this effect as an increase in the expression of inflammatory genes, dysfunction of cardiac cells, and cardiac remodeling [40]. Additionally, Tsutsui et al. reported that various molecular pathways, such as NADPH oxidase, are possibly responsible for ROS production in cardiomyocytes [41]. Jian-Ma et al. reported that Rac1, an important subunit of NADPH oxidase, contributes to doxorubicin-induced cardiac toxicity in cardiomyocytes through ROS-dependent and ROS-independent (HDAC/p53) mechanisms. They considered Rac1 inhibition to be a potentially useful treatment for doxorubicin-induced cardiac toxicity [42]. Henninger et al. mentioned Rho GTPase Rac1 as an essential factor in the regulation of NADPH oxidase, as well as it being the regulator of topoisomerase 2, and showed that these factors play a critical role in the pathophysiology of congestive heart failure (CHF) induced by anthracyclines [43]. Dong et al. observed that quercetin reduced doxorubicin-induced apoptosis, mitochondrial dysfunction, ROS production, and DNA breakdown in H9c2 cells. They also demonstrated an increase in Bcl-2 expression and a decrease in Nox1 expression, which was similar to the results of the present study. They reported that the protective role of quercetin may be associated with NADPH oxidase inhibition, especially NOX1 expression and ROS production [44]. Sirtuin-3 (SIRT3) is a class of mitochondrial lysine deacetylase that regulate mitochondrial respiration and oxidative stress-resistant enzymes such as superoxide dismutase-2. In line with our findings, Cheung et al. reported that doxorubicin decreased the cardiac expression of SIRT3 in mice. They also found that doxorubicin decreased the expression of this gene along with the SOD2 enzyme in H9c2 cardiomyocytes in a dose-dependent manner. Cheung et al. revealed that the overexpression of SIRT3 resulted in increased cardiolipin while mitochondrial respiration was maintained and SOD2 expression was enhanced in doxorubicin-exposed cardiomyocytes. Cheung and colleagues exhibited that the activation of SIRT3 has therapeutic potential for doxorubicin-induced heart failure [45]. Du et al. observed the mRNA and protein levels of a Bcl-2-like protein with a molecular weight of 19 kDa called Bcl-2 interacting protein 3 (Bnip3) increased. Such an increase could be inhibited by the overexpression of SIRT3. They reported that anti-Bnip3 interventions may contribute to the beneficial effect of SIRT3 in preventing mitochondrial damage and heart failure in cancer patients who underwent chemotherapy [46].

### 3.6. Histopathological Evaluations

In this section, the histopathological aspects of the intervention were analyzed by HE staining Figure 7). The histopathologic effects of doxorubicin included myocytolysis, hypertrophy, cell atrophy, discoloration of cell fibers, and undetectable intercalated disks. These detrimental effects can be mitigated by liposomal drug administration, co-administration of quercetin, or a combination of both. Of note, the interventional procedures in this study did not result in a complete reversal of tissue damage. Accordingly, Shaker et al. have suggested that tissue damage may not be completely reversed, since it may require more cardioprotective substances that can protect the cells and structure of the heart [35]. Saad et al. observed that doxorubicin can lead to myocytolysis and myocardial necrosis [47]. Tsutsui et al. stated that ROS could cause structural damage to the myocardium, which leads to myocyte hypertrophy, apoptosis, and interstitial fibrosis by activating MMPs [41]. Yang et al. showed that encapsulating doxorubicin not only increased the safety of the drug by keeping body weight stable and reducing damage to other tissues, based on histological results, but also enhanced the anti-tumor properties of the drug [48]. Nazmi et al. demonstrated an improvement in side effects such as dilation of myofibrils and loss of integrity with quercetin pre-treatment and, like the present study, showed the protective role of quercetin in the hearts of rats [39].

### 3.7. Cellular Analysis, Uptake, and Cytotoxicity

This study was conducted with the aim of reducing the side effects of doxorubicin on cardiac tissues in an animal model as an in vivo test, and also on the H9c2 rat cardiomyoblast cell line as an in vitro analysis. The cellular uptake assay was performed using fluorescence microscopy to evaluate the cell uptake behavior of empty liposomes, free doxorubicin, and liposomal doxorubicin in H9c2 cells after 3 h. The successful delivery of DiL-labeled liposomes to cells is shown in Figure 8. The red and green fluorescence shown in the cell uptake images indicates the entry of doxorubicin and quercetin into the cell, respectively.

It has been shown that free doxorubicin tends to accumulate in the cell nucleus, but its liposomal form also shows its preference in the cytoplasm. This situation is also seen in both PLGA and niosome carriers, and the researchers attribute it to the entry of the carrier by another mechanism, such as endocytosis [36,49]. It should be noted that quercetin has shown a greater tendency to be present in the cytoplasm, which has also been seen in the use of niosomes. Hemati et al. believe that endocytosis plays a key role in the entry of the carrier into the cell compared to the free drug that enters the cell through diffusion. They observed lower fluorescence intensity in human foreskin fibroblast (HFF) cells as normal human cells compared to cancer cell lines and attributed this observation to the lower entry of drug carriers into normal cells as a result of the sensitivity of the synthesized carriers to cancer cells [49]. The liposomes synthesized in this study, similarly to the niosomal doxorubicin used in the study by Hemati et al., were sensitive to pH and their release rate was higher in an acidic environment such that they had less effect on healthy cells. To carry out the cytotoxicity assay, different concentrations of quercetin, doxorubicin, and liposomal doxorubicin were utilized to evaluate the viability of H9c2 cells using the MTT method (Figure 9). The effect of empty liposomes was analyzed; however, their toxicity effect was negligible. With results consistent with ours, Chen et al. applied the MTT method to determine the cytotoxicity of their drug and considered the lack of significant cytotoxicity of the synthesized empty liposomes as a reason for their biocompatibility and drug delivery potential [34]. The free and liposomal forms of doxorubicin were compared at a concentration range of 1.25–15 µg mL^−1^, which showed the viability of about 50 to 5%, from low to high concentrations of this drug. The liposomal form, at concentrations below 15 µg mL^−1^, exhibited lower toxicity than the free form; however, such a difference was not statistically significant. Our findings demonstrated that the difference in cytotoxicity of free doxorubicin and doxorubicin+quercetin was significantly different when the concentrations between 1.25 and 10 µg mL^−1^ of doxorubicin were used. Correspondingly, the difference in the cytotoxicity of liposomal doxorubicin and liposomal doxorubicin+quercetin was significantly different only at a concentration of 10 µg mL^−1^ doxorubicin, whereas at a concentration of 15 µg mL^−1^, no significant discrepancy was observed. The manipulation of free drugs, such as their encapsulation or co-administration with quercetin in the free or encapsulated forms, led to a reduction in their cytotoxicity. However, despite these innovative approaches, in some cases, their therapeutic efficacy is not statistically significant, and, in some cases, it depends on the drug dose. Hemati et al. observed that the cytotoxicity of quercetin and doxorubicin in both free and niosomal forms had less toxicity on human foreskin fibroblast (HFF) cells, a normal cell line, compared with cancer cells. Interestingly, they found that niosomal forms exerted higher toxicity against cancer cells in comparison with the free forms of the two compounds [49]. In a study performed on HUVEC cells by Qureshi et al., they showed that the incorporation of doxorubicin into PLGA copolymer resulted in reduced toxicity of doxorubicin on normal cells. Notably, they demonstrated that the encapsulation of doxorubicin and its combination with quercetin had higher detrimental effects on cancer cells compared with the free form of doxorubicin. Qureshi and colleagues attributed the toxicity of doxorubicin to the production of free radicals and the protective role of quercetin in scavenging reactive oxygen species [36]. In a study conducted by Cote et al., they indicated that doxorubicin-induced cardiac toxicity was attenuated in response to polymeric micelles of resveratrol combined with quercetin. Cote et al. examined the impact of different concentrations of quercetin, resveratrol, and doxorubicin on these cells and showed that doxorubicin was able to initiate apoptosis by activating caspase-3 and -7. They observed that the effects of these interventions varied with the concentrations of substances used and the type of cells. Additionally, these interventions increased the caspase activity in human ovarian cancer cells (SKOV-3), whereas they reduced the activity of caspase enzymes in murine cardiomyocytes (H9c2) when used at a ratio of 10:10:1 (resveratrol, quercetin, doxorubicin, respectively). It is worth noting that although they reduced tissue injury in cardiac tissues, they did not completely reverse the detrimental changes that occurred in cardiomyocytes. Cote et al. attributed the difference in the behavior of the two cell types to different genetic susceptibility [16]. Dong and colleagues also showed a protective effect of quercetin on doxorubicin using the MTT assay performed on HCC2 cells. They found that doxorubicin reduced the number of cells in 48 h in a dose-dependent manner, but with 2 h of pre-treatment with quercetin at concentrations of 50 and 100 μM, the decrease in the number of cells was prevented. Dong and colleagues used doxorubicin at a concentration of 5 μM in their study [44].

## 4. Conclusions

Chemotherapeutic agents have side effects that limit their use; thus, researchers are looking for new ways to reduce these side effects. One of the methods is to use nanotechnology and nanocarriers as a strategy to increase drug effects and reduce adverse effects. The anionic nanocarriers synthesized in our study are liposomes with a simple synthesis method and suitable nanoparticle properties with controlled release and sensitivity to pH. In the present study, the cardiac effects of doxorubicin as a chemotherapy drug and its side effects and the use of its liposomal form, as well as quercetin as a natural antioxidant, were investigated in a rat animal model. It was observed that co-administration of liposomal doxorubicin and quercetin could cause fewer side effects than using the free form of doxorubicin, which was evaluated by biochemical analysis, antioxidant capacity, oxidative stress indices, apoptosis markers, and histopathological assessments. The use of liposomal doxorubicin alone or its non-encapsulated use with quercetin also reduced side effects to some extent. The in vitro results showed that either free doxorubicin and quercetin or liposomal doxorubicin and quercetin are able to reduce cardiotoxicity. The entry of free doxorubicin into the cells was determined by cell uptake assay, and it was observed that the use of liposomal doxorubicin and quercetin reduced this action. Furthermore, a decrease in the cytotoxicity of doxorubicin on H9c2 cells was observed when the drug carrier and quercetin were administered together. However, more studies are needed to further explore reducing the side effects of these drugs.

## Figures and Tables

**Figure 1 pharmaceutics-15-01920-f001:**
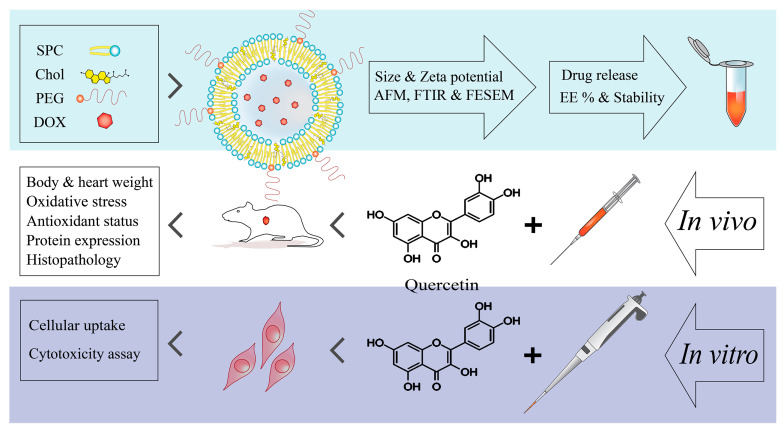
A scheme of study steps.

**Figure 2 pharmaceutics-15-01920-f002:**
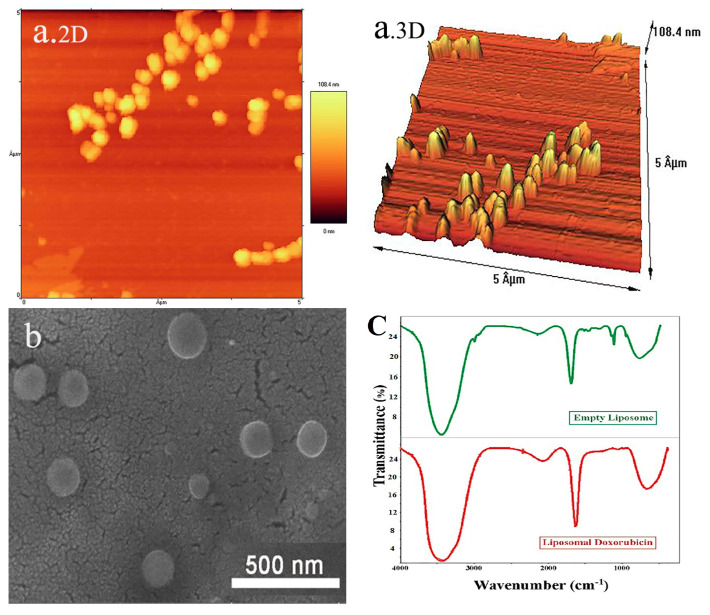
(**a**) AFM images ((**2D**) and (**3D**)) of liposomal doxorubicin. (**b**) Field emission scanning electron microscopy (FE-SEM) of liposomal doxorubicin. (**c**) FTIR spectra of empty liposomes and liposomal doxorubicin.

**Figure 3 pharmaceutics-15-01920-f003:**
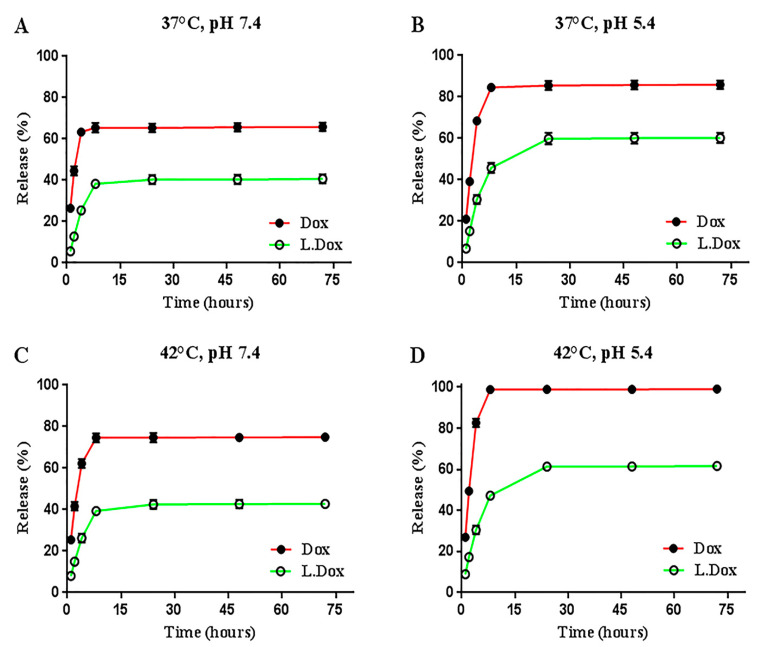
(**A**–**D**) Curves, cumulative release of free and liposomal doxorubicin in different conditions, at temperatures of 37 and 42 degrees Celsius and with pHs of 7.4 and 5.2 after 72 h. Data were presented as mean ± SD (n = 3). Dox: free doxorubicin, L.Dox: liposomal doxorubicin.

**Figure 4 pharmaceutics-15-01920-f004:**
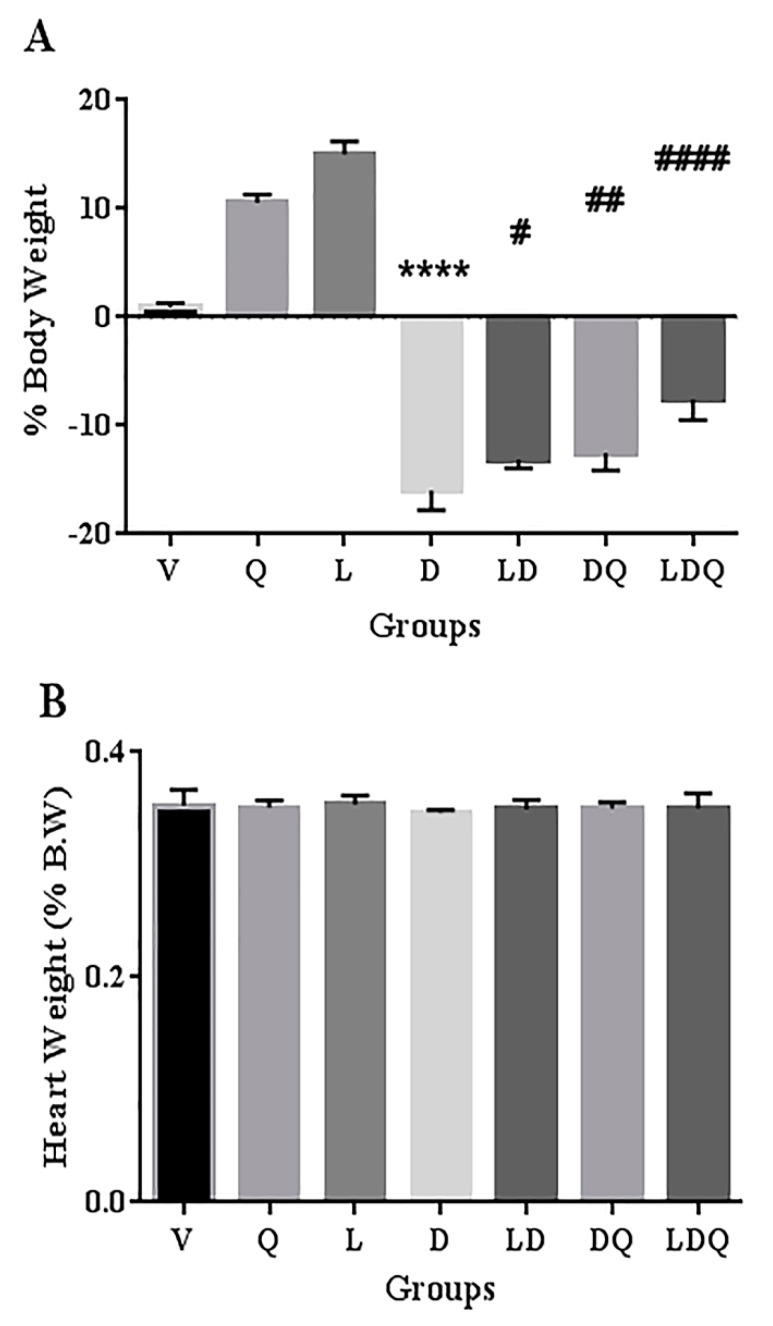
(**A**) Body weight change (%) pre- and post-intervention of different groups (n = 6). (**B**) Heart weight (% body weight) post-intervention of different groups (n = 6). Significant differences between the free drug group (D) and the carrier group (V) are shown as stars: *p* < 0.0001 (****) and significant differences between the free drug group (D) and other drug groups (LD, DQ and LDQ groups) are shown as hashtags: *p* < 0.05 (#), *p* < 0.01 (##), *p* < 0.0001 (####). BW: body weight, V: vehicle, Q: quercetin, L: blank liposome, D: doxorubicin, LD: liposomal doxorubicin, DQ: doxorubicin and quercetin, LDQ: liposomal doxorubicin and quercetin.

**Figure 5 pharmaceutics-15-01920-f005:**
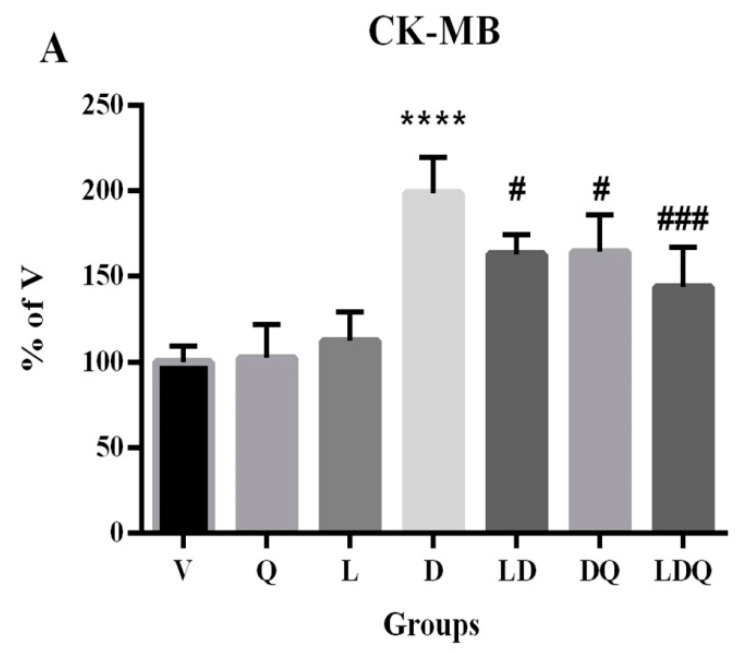
(**A**–**F**) Evaluation of biochemical, antioxidant, and oxidative stress statuses. (**A**,**B**) Serum CK-MB and LDH levels compared to the vehicle group (%); (**C**) MDA content in different groups is expressed as micromoles per milligram of protein; (**D**) GPX activity in different groups is expressed as units per milligram of protein; (**E**) Cat activity in different groups is expressed as units per milligram of protein; (**F**) SOD activity in different groups is expressed as units per milligram of protein. Values are expressed as mean ± SD (n = 6). Significant differences between the free drug group (D) and the carrier group (V) are shown as stars: *p* < 0.05 (*), *p* < 0.0001 (****) and significant differences between free drug group (D) and other drug groups (LD, DQ, and LDQ groups) are shown as hashtags: *p* < 0.05 (#), *p* < 0.01 (##), *p* < 0.001 (###), *p* < 0.0001 (####). V: vehicle, Q: quercetin, L: blank liposome, D: doxorubicin, LD: liposomal doxorubicin, DQ: doxorubicin and quercetin, LDQ: liposomal doxorubicin and quercetin. CK-MB: creatine phosphokinase-MB, LDH: lactate dehydrogenase, GPX: glutathione peroxidase, Cat: catalase, SOD: superoxide dismutase.

**Figure 6 pharmaceutics-15-01920-f006:**
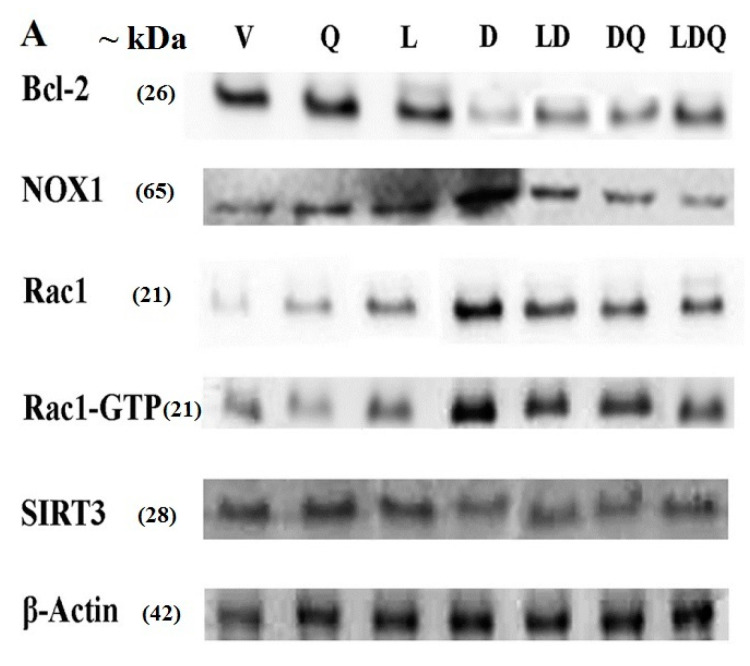
(**A**) Western blot analysis of Bcl-2, NOX1, Rac1, Rac1-GTP, SIRT3 and beta-actin. (**B**) Comparative diagram of Western blot percentage expression for NOX1, Rac1, Rac1-GTP vs beta-actin. (**C**) Comparative diagram of Western blot percentage expression for Bcl-2 and SIRT3 vs beta-actin. Significant differences between the free drug group (D) and the carrier group (V) are shown as a star: *p* < 0.05 (*) and significant differences between the free drug group (D) and other drug groups (LD, DQ and LDQ groups) are shown as a hashtag: *p* < 0.05 (#). V: Vehicle, Q: quercetin, L: Blank liposome, D: doxorubicin, LD: liposomal doxorubicin, DQ: doxorubicin and quercetin, LDQ: liposomal doxorubicin and quercetin.

**Figure 7 pharmaceutics-15-01920-f007:**
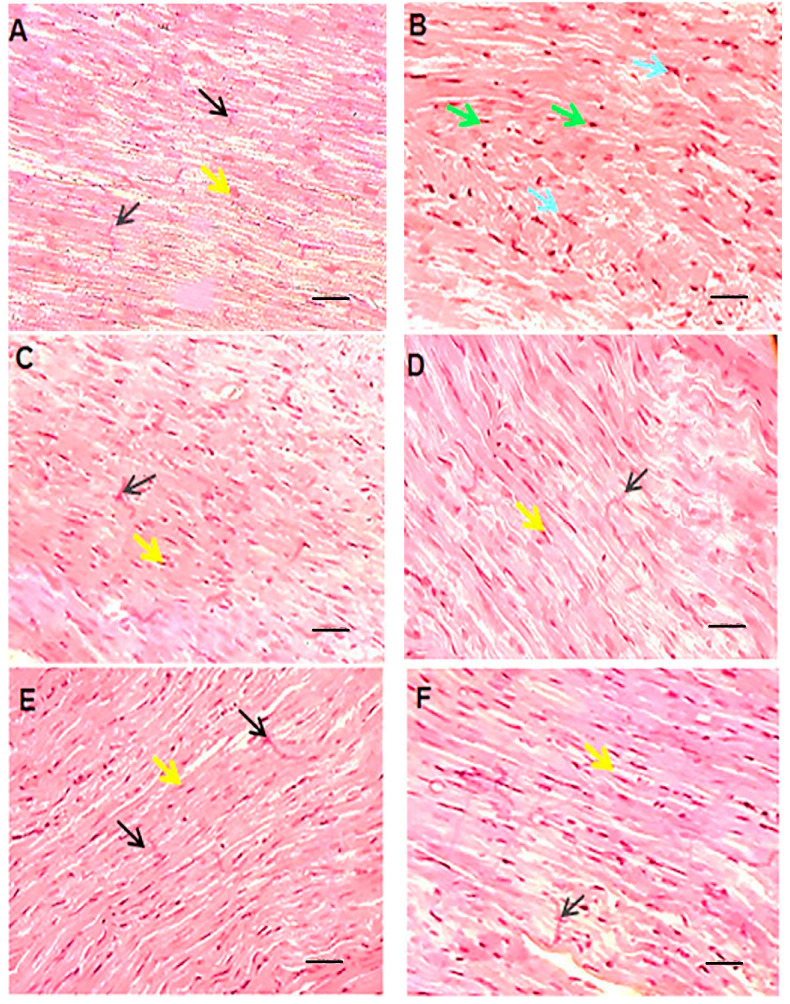
HE staining of left ventricular tissue of intervened-on rats in different groups (scale bar: 100 µm). In the vehicle or control group (**A**), cardiomyocytes are branched-fiber cells connected by intercalated discs (Eberth’s lines) (black arrow). The nuclei are seen as euchromatin, oval, basophilic, and central (yellow arrow). In the free doxorubicin group (**B**), the tissue structure of the cells is seen as myocytolysis, with some cells becoming hypertrophic and some cells atrophic. The nuclei are seen as round and compact (green arrow) or pleomorphic and pyknotic (blue arrow). Empty spaces are created around the nuclei and the margins of the nuclei are seen. The cell fibers are faded and the intercalated discs are indistinguishable. In the empty nanocarrier group and the quercetin group (**C**,**D**), the tissue structure of cardiocytes was normal and without pathological damage similar to the control group. In the groups receiving liposomal doxorubicin, quercetin and doxorubicin, and finally the liposomal doxorubicin and quercetin (**E**–**G**), microscopic findings compared to the doxorubicin group showed fewer tissue irregularities and reduced apoptotic-like cells.

**Figure 8 pharmaceutics-15-01920-f008:**
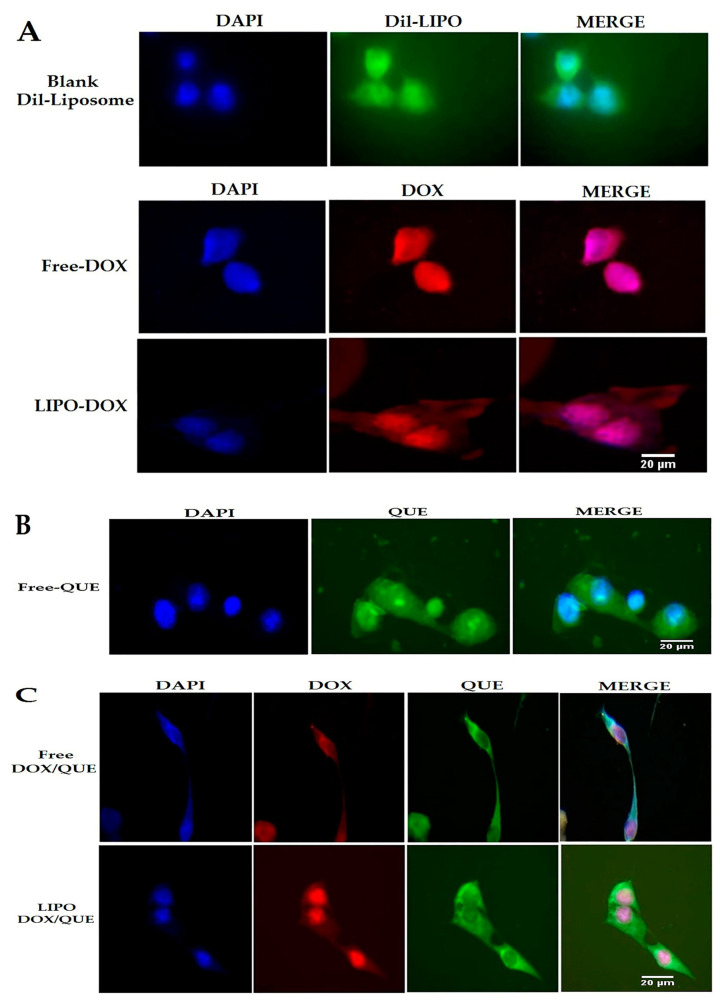
Cellular uptake images of H9c2 cells, incubated with (**A**) DIL-labeled liposome (DIL-LIPO), free doxorubicin (Free-DOX) and liposomal doxorubicin (LIPO-DOX), (**B**) free quercetin (Free-QUE) and (**C**) free doxorubicin and quercetin (Free-DOX/QUE) and liposomal doxorubicin and quercetin (LIPO-DOX/QUE) for 180 min. DAPI (blue) was used for nucleus staining and DIL dye (green) was used for phospholipid staining. QUE (green), DOX (red) (scale bar: 20 µm).

**Figure 9 pharmaceutics-15-01920-f009:**
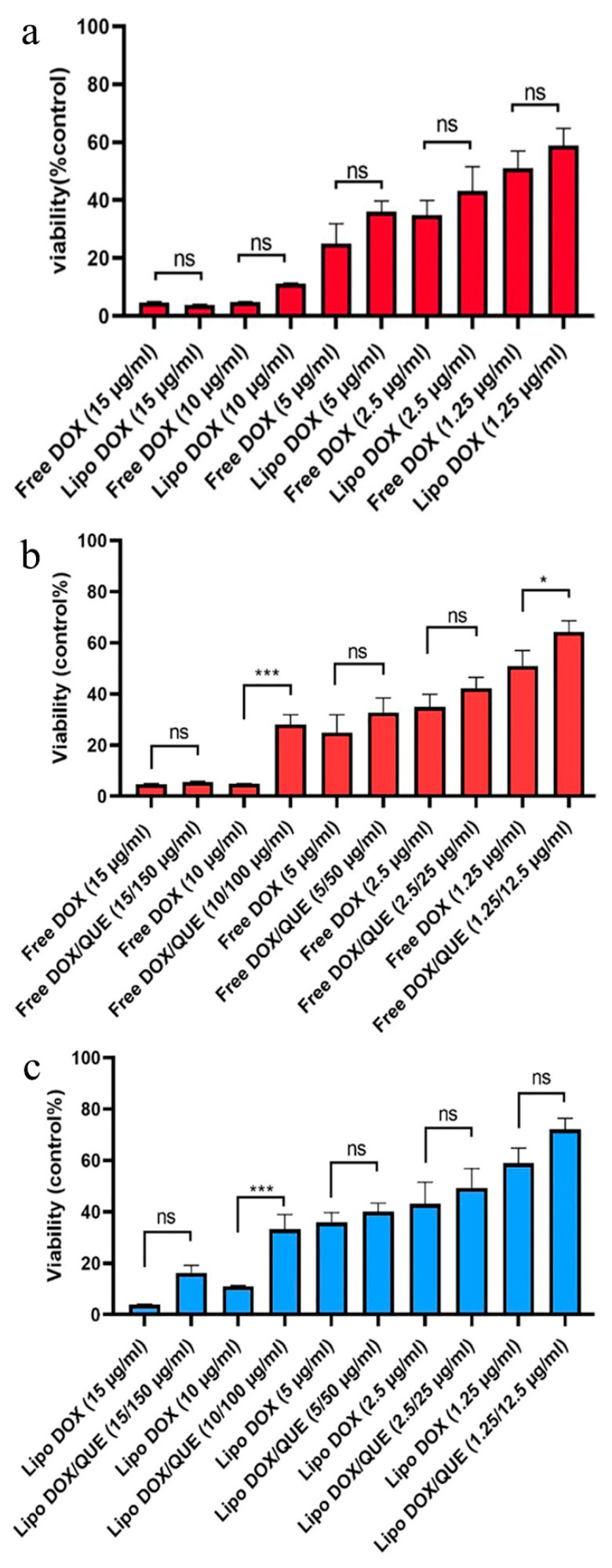
Cytotoxicity assay. (**a**–**c**) Comparison between the toxicity of free doxorubicin, liposomal doxorubicin, free doxorubicin and quercetin, liposomal doxorubicin and quercetin in various concentrations after 48 h for H9c2 cells. Values are expressed as mean ± SD (n = 3). Significant differences are shown as stars: *p* < 0.05 (*) and *p* < 0.001 (***). Abbreviations: not significant (ns), free doxorubicin (free DOX), liposomal doxorubicin (Lipo DOX), free doxorubicin and quercetin (Free DOX/QUE) and liposomal doxorubicin and quercetin (LIPO DOX/QUE).

**Table 1 pharmaceutics-15-01920-t001:** Animal groups and treatments.

Groups	Gavage (Everyday)	Injections (Every Other Day)
V	Normal saline	Phosphate-buffered saline
Q	Quercetin	Phosphate-buffered saline
L	Normal saline	Blank liposome
D	Normal saline	Doxorubicin
LD	Normal saline	Liposomal doxorubicin
DQ	Quercetin	Doxorubicin
LDQ	Quercetin	Liposomal doxorubicin

V: vehicle, Q: quercetin, L: blank liposome, D: doxorubicin, LD: liposomal doxorubicin. DQ: doxorubicin and quercetin, LDQ: liposomal doxorubicin and quercetin, n: 6.

**Table 2 pharmaceutics-15-01920-t002:** Characterization of Liposomes.

Liposome Type	Particle Size (nm)	PDI	Zeta Potential (mV)
Empty liposome	96.3 ± 2.2	0.189 ± 0.02	−19.3 ± 1.5
Liposomal doxorubicin	98.8 ± 2.5	0.204 ± 0.03	−18.1 ± 2.2
Stored liposomal doxorubicin *p*-value	101.9 ± 3.1 0.10	0.222 ± 0.03 0.42	−21.4 ± 2.6 0.24

PDI: Polydispersity index. Data were presented as mean ± SD (n = 3).

## Data Availability

The datasets generated and/or analyzed during the current study are not publicly available due to the university rules and instructions, but are available from the corresponding author upon reasonable request.

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
