# Peer review of "Reduction of Doxorubicin-Induced Cardiotoxicity by Co-Administration of Smart Liposomal Doxorubicin and Free Quercetin: In Vitro and In Vivo Studies"

_pharmaceutics, 2023, doi:10.3390/pharmaceutics15071920_

Round 1

Reviewer 1 Report

In the research article “Reduction of Doxorubicin-Induced Cardiotoxicity by Co-Administration of Smart Liposomal Doxorubicin and Free Quercetin: In Vitro and In Vivo Studies” Dorostkar et al., analyzed the impact of empty liposomes (anionic and pH sensitive), free doxorubicin, liposomal doxorubicin, and quercetin in animal models by measuring cardiac enzymes, oxidative stress and antioxidant 33 markers, protein expression analysis, and histopathological assessments. In addition, the cytotoxicity and cellular uptake were measured in H9c2 cell line. Authors have reported that co-administration of liposomal doxorubicin with free quercetin decreased weight loss, and diminished creatine kinase (CK-MB), lactate dehydrogenase (LDH), and malondialdehyde (MDA), while increasing the activity of glutathione peroxidase (GPX), catalase (CAT), and superoxide dismutase (SOD) enzymes in the left ventricle. Authors have summarized by mentioning that the intervention reported by authors can increase antioxidant capacity, reduce oxidative stress and apoptosis in heart tissue

Comments

2.5. Mention the entire animal protocol as a scheme and list the groups in a Table

2. 8. What percentage gels were used in western blotting?

Authors have to compare the effects of free dox, lipodox along with the effects of Quercetin and check whether the selected combination is yielding a synergistic effect

Figure 6: The western blot data shown in this figure: Did authors analyze the tissues collected from each animal separately or the analysis is a pooled analysis?

Figure 7: Is the data shown in this figure a representative figure? Did authors analyze all the animals in each group?

Minor editing of English language required

Author Response

We would like to thank reviewers for their constructive suggestions and comments. Please find our point-by-point responses below:

Reviewer 1, Comments:
- 2.5. Mention the entire animal protocol as a scheme and list the groups in a Table
A scheme of all the work done in the article as well as a table for better understanding of the animal studies protocol and groupings is included in the article as per your suggestion.

- 2. 8. What percentage gels were used in western blotting?
In the western blot method, the extracted proteins were run on a polyacrylamide gel containing 10% sodium dodecyl sulfate to separate in terms of their molecular weight and then transferred to the nitrocellulose membrane.

- Authors have to compare the effects of free dox, lipodox along with the effects of Quercetin and check whether the selected combination is yielding a synergistic effect.
In this study, the effects of free dox, lipodox and quercetin were compared. Considering that the hypothesis of this study was the possibility of reducing the side effects of doxorubicin by these interventions on healthy rats as well as heart cells, the synergistic effects of these interventions were not investigated. Of course, in future work, we may be able to investigate the effect of these interventions on cancer cell lines and evaluate the synergistic effect.

- Figure 6: The western blot data shown in this figure: Did authors analyze the tissues collected from each animal separately or the analysis is a pooled analysis?
In this figure, the authors analyzed the tissues collected from each animal separately.

- Figure 7: Is the data shown in this figure a representative figure? Did authors analyze all the animals in each group?
In the histopathology studies, the authors analyzed all the animals and the images given in this part of the results are only for observation and comparison between the groups.
--------------------------

Reviewer 2 Report

In this work, Hamideriza developed liposomes for reducing the cardiotoxicity induced by DOX. The cytotoxicity of doxorubicin alone and its encapsulated form in combination with quercetin was evaluated by using different biochemical analyses. This work can be considered for publication after some revisions.

1. The UV-Vis spectra of DOX and its nanoforms should be provided. Also, the standard curve of DOX should be provided.

2.  Why cumulative release of free DOX remains same after 24 h? Why its cumulative release does not reach 100%?

3. Scale bars should be added in Figure 7 and Figure 8.

4. Molecular weight of different proteins should be included in Figure 6A.

Minor English editing is required.

Author Response

We would like to thank reviewers for their constructive suggestions and comments. All changes in the manuscript are highlighted in different colors. Please find our point-by-point responses below:

Reviewer 2, Comments:
1. The UV-Vis spectra of DOX and its nanoforms should be provided. Also, the standard curve of DOX should be provided.
At the beginning of the study, a spectrum of free doxorubicin was prepared to evaluate its approximate optical absorption wavelength to measure the concentration in subsequent experiments. Also, in subsequent studies, the concentration of free doxorubicin was evaluated by the standard curve of free doxorubicin and the concentration of liposomal doxorubicin was also evaluated by the standard curve prepared for liposomal doxorubicin lysed with isopropanol. But due to the presence of many figures in the article, these curves are not included in the article. Of course, if there is a need to present these three curves in the article, this will be done and the graphs will be placed in the article.

2.  Why cumulative release of free DOX remains same after 24 h? Why its cumulative release does not reach 100%?
In most of the articles we reviewed, we observed that only the release of drug systems is investigated and the release of free drugs is not reported. But to show the difference between the release of the nano-drug system and the release of the free drug, we measured the release under different conditions in the two mentioned cases. In the limited articles that have evaluated the release of free dox, different results can be seen. In some of these reports, Dox reaches about 100% release after 24 hours:
Ganassin R, Merker C, Rodrigues MC, Guimarães NF, Sodré CS, Ferreira QD, da Silva SW, Ombredane AS, Joanitti GA, Py-Daniel KR, Zhang J. Nanocapsules for the co-delivery of selol and doxorubicin to breast adenocarcinoma 4T1 cells in vitro. Artificial cells, nanomedicine, and biotechnology. 2018 Nov 17;46(8):2002-12.
Mussi SV, Parekh G, Pattekari P, Levchenko T, Lvov Y, Ferreira LA, Torchilin VP. Improved pharmacokinetics and enhanced tumor growth inhibition using a nanostructured lipid carrier loaded with doxorubicin and modified with a layer-by-layer polyelectrolyte coating. International journal of pharmaceutics. 2015 Nov 10;495(1):186-93.

In some others, the curve has been stopped at less than 100% release, or it is less than 100%.
Wei M, Guo X, Tu L, Zou Q, Li Q, Tang C, Chen B, Xu Y, Wu C. Lactoferrin-modified PEGylated liposomes loaded with doxorubicin for targeting delivery to hepatocellular carcinoma. International journal of nanomedicine. 2015;10:5123.
Diao YY, Li HY, Fu YH, Han M, Hu YL, Jiang HL, Tsutsumi Y, Wei QC, Chen DW, Gao JQ. Doxorubicin-loaded PEG-PCL copolymer micelles enhance cytotoxicity and intracellular accumulation of doxorubicin in adriamycin-resistant tumor cells. International journal of nanomedicine. 2011 Sep 12:1955-62.

And in a study with regard to pH, it is less than 100%, which, of course, is said to have no significant difference:

Biabanikhankahdani R, Bayat S, Ho KL, Alitheen NB, Tan WS. A simple add-and-display method for immobilisation of cancer drug on His-tagged virus-like nanoparticles for controlled drug delivery. Scientific Reports. 2017 Jul 13;7(1):5303.
Finally, we did not see an article that evaluates the release of free DOX in four conditions, two temperatures and two different pH, as well as its nanosystem form, like the present study. Our hypothesis is that free DOX may have a different release under conditions such as temperature and pH, which we saw in the present study, so that at a temperature of 42 degrees and a pH of 5.2, we saw a release of about 100%, but in In other conditions, free drug release was less.
Investigating the causes of the difference in the release of this drug in different conditions requires more investigations, which is beyond the scope of this article and can be further investigated in the future. The issue that was important for us in the present study was the difference between the release of nanosystems and the release of free doxorubicin, which confirms the slow release of the drug from nanoliposomes.

3. Scale bars should be added in Figure 7 and Figure 8.
Thanks, the mentioned item has been corrected.

4. Molecular weight of different proteins should be included in Figure 6A.
Thanks, the mentioned item has been corrected.
--------------------------

Reviewer 3 Report

1.       The layout of the first figure is unfortunate, it takes up an unnecessarily large amount of space. It would be better to put parts A and B next to each other.

2.       Accorcing to FTIR analysis:

What type of thermodynamic changes can be detected by FTIR analysis?

The results of FTIR analysis were confirming the lack of chemical interactions between doxorubicin and liposomes, as well as the stability of doxorubicin within liposomes. If we accept these results as a new result, the reference of 26 and 28 are unnecessary.

Moreover, figure 2 carries so slight visual information that it is sufficient to include it in the supporting material.

3. It is completely pointless to follow the release of the active ingredient in an interval longer than 15 hours. On the other hand, on the initial phase of the release, samples should be taken much more often in order to analyze the kinetics of the release.

Author Response

We would like to thank reviewers for their constructive suggestions and comments. Please find our point-by-point responses below:

Reviewer 3, Comments:
1. The layout of the first figure is unfortunate, it takes up an unnecessarily large amount of space. It would be better to put parts A and B next to each other.
Thanks, the mentioned item has been corrected. Parts a and b were placed next to each other separately.

2.       Accorcing to FTIR analysis:
What type of thermodynamic changes can be detected by FTIR analysis?
The results of FTIR analysis were confirming the lack of chemical interactions between doxorubicin and liposomes, as well as the stability of doxorubicin within liposomes. If we accept these results as a new result, the reference of 26 and 28 are unnecessary. 
In our study, FTIR was used as a method to demonstrate the stability of the drug inside the liposomes. Also, in one of the references of this article, this technique is mentioned as a very useful tool for detecting chemical changes, chemical and thermodynamic structures of phase transition and structural changes in biological systems. If necessary, according to the reviewer's opinion, if this sentence is ambiguous, it can be corrected. 
The reference contains the above content: Al-Rubaie, M. S. & Abdullah, T. S. Multi Lamellar Vesicles (Mlvs) Liposomes Preparation byThin Film Hydration Technique. Engineering and Technology Journal 32, 550-560 (2014).

Moreover, figure 2 carries so slight visual information that it is sufficient to include it in the supporting material.
Our purpose in showing this figure is to compare the diagram of empty and drug-filled liposome, which shows that there are no irrelevant interactions between them. In this edition, we added this figure as a part of the previous figures (Figure 2). According to the reviewer's comments, this figure can be removed.

3. It is completely pointless to follow the release of the active ingredient in an interval longer than 15 hours. On the other hand, on the initial phase of the release, samples should be taken much more often in order to analyze the kinetics of the release.
In this study, the release hours have been done according to the articles that have been published in this regard. For example, in these articles, the release was evaluated at 50 hours, 72 hours, or 100 hours, respectively:
1. Biabanikhankahdani R, Bayat S, Ho KL, Alitheen NB, Tan WS. A simple add-and-display method for immobilisation of cancer drug on His-tagged virus-like nanoparticles for controlled drug delivery. Scientific Reports. 2017 Jul 13;7(1):5303.
2. Abtahi NA, Naghib SM, Ghalekohneh SJ, Mohammadpour Z, Nazari H, Mosavi SM, Gheibihayat SM, Haghiralsadat F, Reza JZ, Doulabi BZ. Multifunctional stimuli-responsive niosomal nanoparticles for co-delivery and co-administration of gene and bioactive compound: In vitro and in vivo studies. Chemical Engineering Journal. 2022 Feb 1;429:132090.
3. Diao YY, Li HY, Fu YH, Han M, Hu YL, Jiang HL, Tsutsumi Y, Wei QC, Chen DW, Gao JQ. Doxorubicin-loaded PEG-PCL copolymer micelles enhance cytotoxicity and intracellular accumulation of doxorubicin in adriamycin-resistant tumor cells. International journal of nanomedicine. 2011 Sep 12:1955-62.
in this study, the release was followed for 72 hours. We evaluated the release of the drug in the first, second, fourth, eighth, 24, 48 and 72 hours, with the aim of investigating the slow release and the effects of changes in the release at different temperatures and pH. If the reviewer wants, the release curve will be displayed only for 24 hours.
--------------------------

Reviewer 4 Report

Authors proposed a research paper entitled “Reduction of Doxorubicin-Induced Cardiotoxicity by Co-Administration of Smart Liposomal Doxorubicin and Free Quercetin: In Vitro and In Vivo Studies” for the publication in Pharmaceutics, MDPI.

 The paper has a good scientific soundness, but the choice of the method of production is obsolete.

 Despite the use of thin layer hydration method for the production of liposomes, the paper has a good scientific soundness, due to the high level of the pharmaceutical application. However, it is necessary to say that Bangham method (the so-called film-hydration method) suffers of several drawbacks, such as solvent residue, post-processing steps needed to reduce mean dimensions and a generally low encapsulation efficiency.

Line 126. “Soy-126 bean phosphatidylcholine (SPC) (Lipoid, Germany),” please specify the commercial name of this phospholipid by Lipoid.

Line 133. “synthesized” I would not talk about synthesis of the film.

Line 133. “55°C for” is the temperature compatible with this kind of phospholipids?

Line 136 “Probe sonication”. As I said, post processing step is needed to obtain nanometric dimensions. How do you guarantee that the encapsulation efficiency is still high?

Line 126. “Probe sonication” what are the probe working conditions ?

Line 138. Since the liposomes are filtered, the Particle Size Distributions are corrected by the filter pore dimensions. What authors saw in the SEM micrograph is the nanometric tail of the PSD.7

Line 311. “Empty liposomes and drug-containing liposomes” the caption is not correct, may “containing dox”.

Drug release diagrams need to be changed. When the curve reaches the plateau, it can be stopped; it is not necessary to provide 2 more 100% points if the release has finished at 15 hours (example in Figure 3d, red lines).

Figure 8. Improve the font of the text in the figure lines.

Line 593. “in vitro” should be written in italics. Correct also in the full manuscript.

A good quality of English. Quite good syntax construction.

Author Response

We would like to thank reviewers for their constructive suggestions and comments. The revisions are highlighted in the revised manuscript.

Please find our point-by-point responses below:

Reviewer 4, Comments:
The paper has a good scientific soundness, but the choice of the method of production is obsolete. Despite the use of thin layer hydration method for the production of liposomes, the paper has a good scientific soundness, due to the high level of the pharmaceutical application. However, it is necessary to say that Bangham method (the so-called film-hydration method) suffers of several drawbacks, such as solvent residue, post-processing steps needed to reduce mean dimensions and a generally low encapsulation efficiency.
Thanks for your encouraging comment. We agree with your opinion about some disadvantages of the film hydration method, but considering that the topic of our study was on liposomes and each method may have advantages and disadvantages, we used this method as the synthesis method. Of course, this method is currently also used as one of the common methods of liposome synthesis. For example, here are some articles that have recently been published in MDPI Journals using this method:
Mureşan, M.; Olteanu, D.; Filip, G.A.; Clichici, S.; Baldea, I.; Jurca, T.; Pallag, A.; Marian, E.; Frum, A.; Gligor, F.G.; Svera, P.; Stancu, B.; Vicaș, L. Comparative Study of the Pharmacological Properties and Biological Effects of Polygonum aviculare L. herba Extract-Entrapped Liposomes versus Quercetin-Entrapped Liposomes on Doxorubicin-Induced Toxicity on HUVECs. Pharmaceutics 2021, 13, 1418. doi: 10.3390/pharmaceutics13091418 
Ferreira-Silva, M.; Faria-Silva, C.; Carvalheiro, M.C.; Simões, S.; Marinho, H.S.; Marcelino, P.; Campos, M.C.; Metselaar, J.M.; Fernandes, E.; Baptista, P.V.; Fernandes, A.R.; Corvo, M.L. Quercetin Liposomal Nanoformulation for Ischemia and Reperfusion Injury Treatment. Pharmaceutics 2022, 14, 104. doi: 10.3390/pharmaceutics14010104 
Zhang, Z.; Ma, L.; Luo, J. Chondroitin Sulfate-Modified Liposomes for Targeted Co-Delivery of Doxorubicin and Retinoic Acid to Suppress Breast Cancer Lung Metastasis. Pharmaceutics 2021, 13, 406. doi: 10.3390/pharmaceutics13030406 
Kabel, A.M.; Salama, S.A.; Adwas, A.A.; Estfanous, R.S. Targeting Oxidative Stress, NLRP3 Inflammasome, and Autophagy by Fraxetin to Combat Doxorubicin-Induced Cardiotoxicity. Pharmaceuticals 2021, 14, 1188. doi: 10.3390/ph14111188 

Line 126. “Soy-126 bean phosphatidylcholine (SPC) (Lipoid, Germany),” please specify the commercial name of this phospholipid by Lipoid. 
Thanks, the mentioned item has been corrected.

Line 133. “synthesized” I would not talk about synthesis of the film.
Was corrected: The film was hydrated at 55 °C for 60 minutes.

Line 133. “55°C for” is the temperature compatible with this kind of phospholipids?
According to similar articles and studies, a temperature of 55 or 60-70 degrees has been used to hydrate the liposomal thin film:
Parchami M, Haghiralsadat F, Sadeghian-Nodoushan F, Hemati M, Shahmohammadi S, Ghasemi N, Sargazi G. A new approach to the development and assessment of doxorubicin-loaded nanoliposomes for the treatment of osteosarcoma in 2D and 3D cell culture systems. Heliyon. 2023 Apr 20.
Andra VV, Pammi SV, Bhatraju LV, Ruddaraju LK. A comprehensive review on novel liposomal methodologies, commercial formulations, clinical trials and patents. Bionanoscience. 2022 Mar;12(1):274-91.
Dhas N, Preetha HS, Dubey A, Ravi G, Govindan I, Rama A, Naha A, Hebbar S. Factorial design-based fabrication of biopolymer-functionalized Asiatic acid-embedded liposomes: in-vitro characterization and evaluation. Journal of Applied Pharmaceutical Science. 2022 Nov 5;12(11):071-81.
Wang WY, Cao YX, Zhou X, Wei B. Delivery of folic acid-modified liposomal curcumin for targeted cervical carcinoma therapy. Drug Design, Development and Therapy. 2019 Jul 4:2205-13.
Haghiralsadat F, Amoabediny G, Sheikhha MH, Zandieh‐doulabi B, Naderinezhad S, Helder MN, Forouzanfar T. New liposomal doxorubicin nanoformulation for osteosarcoma: Drug release kinetic study based on thermo and pH sensitivity. Chemical biology & drug design. 2017 Sep;90(3):368-79.
Haghiralsadat F, Amoabediny G, Naderinezhad S, Zandieh-Doulabi B, Forouzanfar T, Helder MN. Codelivery of doxorubicin and JIP1 siRNA with novel EphA2-targeted PEGylated cationic nanoliposomes to overcome osteosarcoma multidrug resistance. International Journal of Nanomedicine. 2018;13:3853.
Alavizadeh SH, Badiee A, Golmohammadzadeh S, Jaafari MR. The influence of phospholipid on the physicochemical properties and anti-tumor efficacy of liposomes encapsulating cisplatin in mice bearing C26 colon carcinoma. International journal of pharmaceutics. 2014 Oct 1;473(1-2):326-33.

Line 136 “Probe sonication”. As I said, post processing step is needed to obtain nanometric dimensions. How do you guarantee that the encapsulation efficiency is still high?
Of course, a post-processing step is required to obtain nanometer dimensions, and finally, the characterization of the synthesized liposomal system has been done after all the post-processing steps.

Line 126. “Probe sonication” what are the probe working conditions ?
Multilamellar formed vesicles (MLVs) were sonicated to produce small unilamellar vesicles (SUVs) for 60 min using a microtip probe sonicator (E–Chrom Tech Co, Taiwan) equipped with a micro tip with a diameter of 5 mm and a power of 60 W inside an ice bath.
Parchami M, Haghiralsadat F, Sadeghian-Nodoushan F, Hemati M, Shahmohammadi S, Ghasemi N, Sargazi G. A new approach to the development and assessment of doxorubicin-loaded nanoliposomes for the treatment of osteosarcoma in 2D and 3D cell culture systems. Heliyon. 2023 Apr 20.

Line 138. Since the liposomes are filtered, the Particle Size Distributions are corrected by the filter pore dimensions. What authors saw in the SEM micrograph is the nanometric tail of the PSD.7
Of course, liposomes are filtered and the particle size distribution is modified by the filter pore dimensions. Our goal in preparing the electron microscope image was to observe nano-sized liposomes and see their morphology.

Line 311. “Empty liposomes and drug-containing liposomes” the caption is not correct, may “containing dox”.
Thanks, the mentioned item has been corrected. It was replaced by the term liposomal Doxorubicin.

Drug release diagrams need to be changed. When the curve reaches the plateau, it can be stopped; it is not necessary to provide 2 more 100% points if the release has finished at 15 hours (example in Figure 3d, red lines).
Before drawing the release diagram, we did not know about the release method and according to the available articles, we continued this study until 72 hours to see the possible increase of the release in the next hours and to be able to compare it in different situations. Of course, if the reviewer wants, for example, the release curve can be shown only up to 24 hours.

Figure 8. Improve the font of the text in the figure lines.
Thanks, the mentioned item has been corrected.

Line 593. “in vitro” should be written in italics. Correct also in the full manuscript.
We sincerely thank the reviewer for constructive suggestion. The mentioned item has been corrected as directed.

_____________________________________________

Round 2

Reviewer 2 Report

I agree to accept it for publish.

Reviewer 4 Report

Authors reponded point by point to my issues.

The quality of the paper improved according to th variations performed by the authors.

Nothing more to declare.

The use of English is clear.